# Longitudinal Assessment of Tau-Associated Pathology by ^18^F-THK5351 PET Imaging: A Histological, Biochemical, and Behavioral Study

**DOI:** 10.3390/diagnostics11101874

**Published:** 2021-10-12

**Authors:** Ines Moreno-Gonzalez, George A. Edwards, Omar Hasan, Nazaret Gamez, Jonathan E. Schulz, Juan Jose Fernandez-Valenzuela, Antonia Gutierrez, Claudio Soto, Paul E. Schulz

**Affiliations:** 1Department of Neurology, The University of Texas Health Science Center at Houston, Houston, TX 77030, USA; George.Edwards@bcm.edu (G.A.E.III); Omar.Hasan@uth.tmc.edu (O.H.); Nazaret.GamezRuiz@uth.tmc.edu (N.G.); jonathan.schulz@uth.tmc.edu (J.E.S.); Claudio.Soto@uth.tmc.edu (C.S.); 2Department of Cell Biology, Instituto de Investigacion Biomedica de Malaga-IBIMA, Faculty of Sicences, University of Malaga, 29010 Malaga, Spain; juanjofv@uma.es (J.J.F.-V.); agutierrez@uma.es (A.G.); 3Networking Biomedical Research Center on Neurodegenerative Diseases (CIBERBED), 28031 Madrid, Spain

**Keywords:** PET imaging, Tau, THK5351, tracer, biomarker, histological assessment, cognitive impairment, tauopathy, Alzheimer’s disease, dementia

## Abstract

Several common and debilitating neurodegenerative disorders are characterized by the intracellular accumulation of neurofibrillary tangles (NFTs), which are composed of hyperphosphorylated tau protein. In Alzheimer’s disease (AD), NFTs are accompanied by extracellular amyloid-beta (Aβ), but primary tauopathy disorders are marked by the accumulation of tau protein alone, including forms of frontotemporal dementia (FTD), corticobasal degeneration (CBD), and progressive supranuclear palsy (PSP), among others. ^18^F-THK5351 has been reported to bind pathological tau as well as associated reactive astrogliosis. The goal of this study was to validate the ability of the PET tracer ^18^F-THK5351 to detect early changes in tau-related pathology and its relation to other pathological hallmarks. We demonstrated elevated in vivo ^18^F-THK5351 PET signaling over time in transgenic P301S tau mice from 8 months that had a positive correlation with histological and biochemical tau changes, as well as motor, memory, and learning impairment. This study indicates that ^18^F-THK5351 may help fill a critical need to develop PET imaging tracers that detect aberrant tau aggregation and related neuropathology in order to diagnose the onset of tauopathies, gain insights into their underlying pathophysiologies, and to have a reliable biomarker to follow during treatment trials.

## 1. Introduction

Tau protein aggregation and deposition occurs in several neurodegenerative dementias that are collectively called tauopathies. The most common cause of dementia is Alzheimer’s disease (AD), which affects about 47 million individuals worldwide and accounts for 60–80% of dementia cases [1]. AD is an age-related, progressive dementia characterized by extracellular amyloid-beta (Aβ) plaques, intracellular neurofibrillary tangles (NFTs) composed of hyperphosphorylated tau protein, inflammation, synaptic impairment, and neuronal loss. These result in macroscopic brain atrophy and cognitive function demise. Amyloid deposition can be detected up to 15–20 years before the onset of symptoms [2]. According to the amyloid cascade hypothesis, Aβ aggregation is followed by hyperphosphorylated p-tau protein deposition [3]. Other dementias have p-tau deposition, and include frontotemporal dementia (FTD), progressive supranuclear palsy (PSP), corticobasal degeneration (CBD), chronic traumatic encephalopathy (CTE), primary age-related tauopathy (PART), argyrophilic grain disease (AGD), and globular glial tauopathy (GGT) [4].

Tau protein is encoded by the microtubule-associated protein tau gene (MAPT), which is located on chromosome 17q21, and expressed throughout the brain, chiefly in neuronal cells. In the adult human brain, there are six isoforms of tau that are created by alternative splicing and correspond to the inclusion of an additional microtubule-binding repeat region (3R or 4R) at the C-terminal and insertion of 29 amino acids at the N-terminal (0N, 1N, or 2N), varying in length from 352 to 441 amino acids [5]. Physiologically, 3R and 4R are similar in the human brain, yet 0N and 1N are in the majority. Importantly, different tauopathies tend to include specific isoforms. For example, they can have 4R only (PSP, CBD, AGD, GGT); 3R only, (FTD); or mixed 3R/4R (AD, CTE, Down syndrome) [6].

Tau protein dynamically binds to microtubules to promote cytoskeletal assembly and stability, thereby aiding in axonal transport, as well as establishment and maintenance of neuronal polarity. It is well regulated by post-translational modifications, including phosphorylation and dephosphorylation. Phosphorylation from several protein kinases reduces the propensity for binding to microtubules, thereby altering stability and shifting to disassembly, whereas dephosphorylation ameliorates this effect [7]. Phosphorylation of tau has been reported at up to 85 specific sites. In disease states, a hyperphosphorylation of tau occurs that releases tau from the microtubule, constituting the major protein subunit of paired helical filaments (PHF) and NFTs. During the misfolding and polymerization process, tau protein undergoes various secondary conformational changes as it transitions from oligomers to protofibrils to insoluble fibrils. Accumulation of NFTs is suggested to be a fundamental driver in tauopathies [8,9]. Tauopathies may differ in clinical symptoms and onset, brain regions with pathological tau, type and isoform of tau aggregates, and ultrastructural and biochemical properties. Moreover, tau aggregation correlates better with neuronal loss and clinical symptoms than Aβ in AD [10,11], suggesting a direct pathologic role for tau accumulation.

Tau aggregates have been shown to spread following a spatiotemporal pattern, as exemplified by Braak staging in AD. Braak and Braak describe the pathological stages of tau in AD as following a stereotypical and temporally distinct pattern of deposition emulating areas synaptically connected. In Braak I-II stages NFTs first develop in the trans-entorhinal cortex and locus coeruleus; in III-IV stages, the previous affected areas have more robust tau deposition and NFTs are located in the hippocampus, fusiform gyrus, and temporal cortex; and in the final V-VI stages, the NFTs are abundant and spread throughout the cortex [12,13].

Neuroimaging could be a significant tool to detect brain abnormalities and pathological alterations related to tau. It is helpful in analysis of chronic progressive alterations such as changes in cortical thickness, ventricle dilatation, and brain atrophy by computed tomography (CT) and magnetic resonance imaging (MRI). Recent advances in the field of positron emission tomography (PET) likely offer great potential for identifying progression of neurodegeneration. Tau PET imaging represents a great opportunity to detect pathological tau in vivo in specific brain regions to study its neuropathological spread, to identify tauopathies, and to monitor progression of disease. Tau PET imaging would also be less invasive for longitudinal studies than other methods, such as obtaining cerebrospinal fluid (CSF). Although there are several radioligands approved to detect Aβ deposition, such as ^18^F-Florbetapir (Amyvid), there is only one approved for detecting tau pathology, i.e., ^18^F-Flortaucipir (Tauvid), but its specificity is unclear. A new set of radioligands to detect tangles has been reported to selective binding to tau in AD patients and animal models of AD, including THK5351, ^18^F-T808, ^18^F-PBB3, MK-6240, and RO948 [14].

Using ^18^F-THK5117, Ishiki et al. followed AD and age-matched healthy control patients up to 1.5 years and demonstrated yearly tau burden increases in temporal cortex and the fusiform gyrus, as well as widespread distribution throughout cortical regions compared to control patients. Moreover, they correlated with cognitive deficits [15]. There is also a great need to test tau PET imaging agents in non-AD tauopathies. ^18^F-THK5351 has been reported to bind pathological tau as well as associated reactive astrogliosis [16,17].

While non-invasive imaging to detect and track tau pathology in at-risk individuals has great promise, tracers need to be validated. The goal of this study was to test the hypothesis that the PET tracer ^18^F-THK5351 can identify and track the longitudinal progression and distribution of tau aggregation and related pathology in the brain that will correlate with molecular brain pathology and clinical signs. To accomplish this, we used the tau transgenic mouse model P301S (PS19 strain) and compared longitudinal PET scanning results to histology and biochemistry, as well as behavioral changes. Our results revealed a significant age-dependent accumulation of tau pathology throughout the brain concomitant with motor and cognitive impairments in the P301S mice. These results support the development of tau PET imaging as a diagnostic approach for various tauopathies that will aide us in better understanding the pathophysiology underlying tauopathies, in combination with other tracers and clinical tests, and may serve as a biomarker to follow treatment efficacy.

## 2. Materials and Methods

### 2.1. Animals

P301S transgenic mice (Prnp-MAPT*P301S P301SVle/J, Jackson Laboratory, Bar Harbor, ME, USA) express the P301S mutant human microtubule-associated protein tau (MAPT) under the control of the mouse prion protein (Prnp) promoter. These animals accumulate hyperphosphorylated tau that can be detected from 6 months of age, as described previously [18]. The expression of the mutant human MAPT is fivefold higher than the expression of the endogenous mouse MAPT. Animals were housed in groups of up to 5 in individually ventilated cages under standard conditions (22 °C, 12 h light–dark cycle) receiving food and water ad libitum. All animal experiments were carried out in accordance with the NIH regulations and approved by the committee of animal use for research at the University of Texas Health Science Center at Houston, McGovern Medical School. During PET imaging evaluation, animals were transported to the University of Texas MD Anderson Cancer Center (MDACC). Animals were housed per agreements with the DVMS Laboratory Animal Medicine personnel before transportation to MDACC. Radioactive agents for animal use, restraint, and anesthesia use were approved by IACUC. All mice were manipulated in accordance with UT MD Anderson’s IACUC guidelines.

### 2.2. Rotarod

Motor coordination and activity, grip strength, and fatigue are all measurements that are assessed by the commonly used rotarod behavioral task (Med Associates Inc., Fairfax, VT, USA). As previously described [19,20], each animal was placed on a horizontally oriented, rotating cylinder that accelerated from 4 to 40 rpms over a 5 min trial. The latency to fall was quantified and averaged over 3 trials with a 2 min rest interval. The final cut-off latency to fall measurement was 6 min. To prevent false interpretations or falls, we considered two consecutive rotations of the animal clinging to the rod as a latency to fall measurement. The animals were aided for 30 s in the first trial to prevent unnecessary behavior (i.e., turning and false-positive falls/jumps). Animals were blindly assessed longitudinally at 6 months, 8 months, 10 months, and 12 months of age after brain imaging.

### 2.3. Barnes Maze

The Barnes maze was performed as previously reported [21,22]. The Barnes maze is a medial temporal lobe-dependent task that evaluates spatial learning and memory. The Barnes maze utilized here consisted of a circular platform with 40 holes, with one being an exit from the arena. The animal uses spatial cues or shapes to discover the stationary escape hole in a 3 min trial. To run Barnes maze as a longitudinal behavioral task, we trained mice on day 1 with two additional acquisition trials and then 2 trials per day for 4 days. This training phase was performed in a subset of mice at 6 months and 8 months of age to establish an age-dependent performance in the training phase between transgenic and WT mice. Thereafter, two additional trials to assess memory recall were performed at 10 and 12 months of age prior to brain PET imaging. Primary latency to the escape hole was used to assess learning and memory. Task performance was recorded and analyzed using the TopScan 2.0 tracking software.

### 2.4. Longitudinal PET Imaging with ^18^F-THK5351 Tau Tracer

We acquired 10 mCi of radioactive tracer ^18^F-THK5351 (≥95% purity) from the Cyclotron Radiochemistry Facility at University of Texas MD Anderson Cancer Center for each imaging session. Animals were imaged in the Small Animal Imaging Facility (SAIF) at The Center for Advanced Biomedical Imaging (CABI), The University of Texas MD Anderson Cancer Center. The final product of ^18^F-THK5351 injection consisted of ^18^F-THK5351 in 9% ethanol in saline solution: ^18^F-THK5351 (71.5 mCi to 572 mCi); 0.9% sodium chloride for injection USP (13.0 mL); and ethanol 200 proof, USP (1.3 mL). The ^18^F-THK5351 precursor was THK-5352 from Tohoku University. Mice were anesthetized using 2% isoflurane, and a catheter was placed in the tail vein (*n* = 10–11/group). The anesthetized mice were then positioned on a Bruker Albira PET/SPECT/CT scanner (Bruker Biospin Corp., Billerica). A bolus injection of 150 µCi of ^18^F-THK5351 (an average of 5.75 ± 0.44 µCi per gram of body weight) was given via the catheter in the tail vein at the start of the PET measurement followed by a saline flush. During standardization of the probe, dynamic imaging was performed over 30 min every 20 s. Activity plateau was reached between 15 and 20 min (Appendix A). Therefore, the imaging protocol consisted of a 20 min PET followed by a 3 min CT scan (400 uA, 45 kV, 120 projections). The PET data were acquired in list mode and were histogrammed into 22 timeframes (15 × 20 s, 5 × 60 s, 2 × 300 s). The images were reconstructed using the maximum likelihood expectation maximization (MLEM) method with 12 iterations. Scatter, random coincidences, decay, and attenuation corrections were applied. Analysis was conducted with PMOD (PMOD Technologies Ltd., Zürich, Switzerland). The mean standardized uptake value ratio per body weight (SUV-bw) (g/mL) of the brain region of interest (ROI) and normal muscle tissue were measured over the course of the scan, and a ratio of the brain to muscle was calculated.

Brain ROI-based PET data were acquired in list mode and plotted into timeframes. Averaged tracer retention volume, as the mean standardized uptake value ratio per body weight (SUV-bw) (g/mL) of the ROI, was measured over the course of the scan in several ROIs (Appendix A). Selected ROIs included striatum (STR), cortex (CTX), hippocampus (HP), thalamus (TH), cerebellum (CB), basal forebrain and septum (BFS), hypothalamus (HY), amygdala (AMY), brain stem (BS), cingulate gyrus (CG), superior colliculi (SC), olfactory bulb (OF), midbrain (MID), and inferior colliculi (IC).

### 2.5. Histology for Tau Pathology

Animals for histological assessment (P301S and WT) were analyzed (*n* = 4/age/group) at 6 months, 8 months, 10 months, and 12 months of age. Since PET imaging was performed longitudinally, only 12-month-old animals were the same individuals as the ones used for imaging. Mice were sacrificed by CO_2_ inhalation and were perfused trans-cardially with PBS-EDTA. Brains were removed, post-fixed into 10% neutral buffered formalin fixative solution, and embedded in paraffin. Paraffin-fixed brain tissue was sliced coronally by a microtome obtaining 10 μm thick serial sections that were processed for immunostaining. Four distinct coronal sections were assayed per slide (1 slide every 10th section), as performed previously. The endogenous peroxidase activity was blocked with 3% H_2_O_2_-10%, 10% methanol in PBS for 20 min. Then, sections were incubated overnight at room temperature in either monoclonal AT8 phospho-tau pSer202+Thr205 antibody (1:100, Thermo Fischer Scientific, Waltham, MA, USA) or MC1, which analyzes pre-tangle PHFs (a gift from Peter Davies, 1:350). Primary antibody was detected by incubating 2 h with sheep anti-mouse HRP-linked secondary antibody (GE Healthcare Bio-Sciences Corp, Piscataway, NJ, USA), and peroxidase reaction was visualized using DAB Kit (Vector) following the manufacturer’s instructions. Finally, all sections were dehydrated in graded ethanol, cleared in xylene, and cover slipped with DPX mounting medium. Gallyas silver staining detects argentophilic inclusion bodies, such as NFTs, and was performed as previously described [23]. Briefly, tissue was incubated in 0.25% potassium permanganate for 15 min, followed by 2% oxalic acid for 2 min and 5% periodic acid for 7 min. After washing, slices were incubated in silver iodide solution and acetic acid, and then they were developed with Gallyas working solution (anhydrous sodium carbonate, ammonium nitrate, silver nitrate, and tungstosilicic acid). Finally, samples were dehydrated and covered with DPX.

### 2.6. Histological Image Analysis and Quantification

Image attainment was performed by whole slide scanning by Pathscan Enabler IV (Meyer Instruments) for quick overall AT8, MC1, and Gallyas brain imaging and quantification. Images were then imported into ImageJ 1.45s software (NIH), where they were converted to black and white images. Threshold intensity was used to quantify the histological burden in the whole brain. Threshold intensity was automatically adjusted to remove the background and was held constant during quantification of all the analyzed animals. Burden was defined as the immunoreactive area per total area analyzed in percentage. Then, slides were examined using bright field microscopy (DMI6000B, Leica Microsystems, Buffalo Grove, IL, USA), and photomicrographs were taken with a digital camera (DFC310 FX Leica Microsystems, Buffalo Grove, IL, USA).

### 2.7. ELISA Quantification

Brains from P301S mice at 6 months, 8 months, 10 months, and 12 months of age (*n* = 5–7/age) were homogenized at 10% weight/volume in PBS containing protease inhibitors, as previously described [22]. Brain homogenates were centrifuged at 32,600 rpm for 1 h at 4 °C in an ultracentrifuge (Beckman-Coulter, Brea, CA, USA). The supernatant was removed, and pellets were resuspended in 200 μL of 70% formic acid followed by sonication. Samples were centrifuged for 30 min under the same conditions, and the supernatant was collected and neutralized in 1 M Tris-HCl buffer, pH 10.8, to measure the fraction of insoluble tau in brain. Levels of phosphorylated tau at Ser199 were measured using Human tau (pS199) ELISA kit (Invitrogen, Waltham, MA, USA) per the manufacturer’s instructions on an ELISA plate reader (EL800 BioTek, Winooski, VT, USA).

### 2.8. Statistical Analysis

After confirming a normal distribution with the Skewness–Kurtosis statistic test, we used a one-way analysis of variance (ANOVA) followed by a post hoc Tukey’s multiple comparisons test to analyze differences between groups. Two-way ANOVA followed by Tukey’s multiple comparisons test was used for behavior data. Statistical analyses were performed using GraphPad Prism 8.2 software (GraphPad Software, San Diego, CA, USA). Statistical differences for all tests were considered significant at the *p* < 0.05 level.

## 3. Results

### 3.1. Longitudinal PET Imaging Showed Augmented Pathological Tau in P301S Mice

P301S and WT animals underwent longitudinal brain PET/CT imaging using the ¹⁸F-THK5351 radiotracer that recognizes tau aggregates at 6, 8, 10, and 12 months of age. The ^18^F-THK5351 tau tracer was imaged, and the summed dynamic emission frames were registered. We found that the tau radiotracer was able to reach the brain and cross the blood–brain barrier, binding to the cortical and brain stem areas in P301S animals stronger than in WT controls.

Figure 1A shows representative PET imaging images in 6-month-old P301S mice and age-matched WT mice. PET imaging was overlaid with the CT skull imaging and maximum intensity projection (MIP). Figure 1B shows example PET images at 8 months of age, illustrating a larger overall retention of the radiotracer in brains of the P301S mice as compared to WT littermate rodents. Figure 1C shows PET images at 10 months of age, following behavioral assessment, and demonstrates more robust pathological tau in P301S animals as compared to WT mice. Finally, Figure 1D shows the same P301S and WT animals that were re-imaged at 12 months of age. There was a drop in the number of rodents, however, mostly due to onset of paralysis or spontaneous death. Despite this, the remaining P301S animals had elevated tau tracer uptake in PET imaging as compared to age-matched WT animals.

Brain region of interest (ROI)-based PET data were acquired, and averaged tracer retention volume, as the mean standardized uptake value ratio per body weight (SUV-bw) (g/mL), was measured over the course of the scan in several ROIs. We analyzed the striatum, cortex, hippocampus, thalamus, cerebellum, basal forebrain and septum hypothalamus, amygdala, brain stem, cingulate gyrus, superior colliculi, olfactory bulb, midbrain, and inferior colliculi. Statistical parametric mapping was analyzed (Figure 2).

At 6 months, there was no significant difference by two-way ANOVA analysis of the averaged SUV in the overall brain (F(1, 252) = 0.2831, *p* = 0.5952) nor in individual brain regions as per the post hoc Tukey test (Figure 2A). Statistical analysis of the PET imaged brain at 8 months demonstrated a significant difference in the overall brain between P301S and WT mice (Figure 2B, two-way ANOVA, F(1, 225) = 13.90 *p* = 0.0002). Post hoc analysis by Tukey test showed that there was no significant difference between brain regions (Figure 2B). When comparing the averaged SUV rate of the animals at 10 months of age, we found that P301S exhibited significant difference in tau PET imaging in the brain as compared to WT mice (Figure 2C, F(1, 140) = 23.09, *p* < 0.0001). When assessing the average SUV in the brain of the ^18^F-THK5351 radioligand at 12 months of age, we found that two-way ANOVA analysis demonstrated a significant difference between transgenic tau mice and WT littermate (Figure 2D, F(1, 70) = 18.30, *p* < 0.0001). There was no distinct difference when assaying the ROIs by post hoc analysis. Overall, longitudinal tau PET imaging in P301S and WT mice demonstrated a higher uptake of the tau tracer ^18^F-THK5351 in the brain most prominently from 8 months to 12 months of age in P301S animals. This increase was higher in OF compared to CB over time.

### 3.2. P301S Mice Exhibited Longitudinal Motor, Learning, and Memory Impairments

To determine whether results obtained in PET imaging correlated with clinical signs observed in the animal model, we evaluated transgenic tau P301S and age-matched WT littermate animals who underwent longitudinal tau PET imaging for longitudinal motor and behavioral performance at 6 months, 8 months, 10 months, and 12 months of age, right before imaging.

Motor function was examined longitudinally by rotarod testing at 6, 8, 10, and 12 months of age in P301S and WT littermate mice. Whereas there was no significant difference noted at 6 months of age compared to WT (*p* > 0.05), significant motor impairment in the P301S animals was present at 8 months (*p* < 0.05), 10 months (*p* < 0.01), and 12 months of age (*p* < 0.05) in comparison with age-matched WT mice (Figure 3A). These differences were also significant when comparing P301S mice at 6 months with the same animal at 8 (*p* < 0.05) and 10 (*p* < 0.01) months (mixed-effect model). Motor impairment is a known phenotypic characteristic for the P301S mouse model [18,24], where animals display retracted hindlimbs and paralysis. This progressive paralysis occurred, beginning around 8 months of age, and was directly correlated with an approximately 20% survival at 12 months, as observed in the survival curve (Figure 3B).

At age 6 months, there was no significant difference in the learning curve on the Barnes maze between P301S and WT mice (Figure 3C; *p* > 0.05, two-way ANOVA, F(1, 7) = 0.53). At 8 months, however, P301S mice displayed a significant difference discovering the escape hole versus WT mice (Figure 3D; *p* < 0.0001, two-way ANOVA, F(1, 10) = 34.02). Both groups displayed a significant difference as a function of time at the 6- and 8-month age time-points (*p* < 0.0001 and *p* < 0.001, respectively). After the training trials, two additional Barnes maze trials were run at the 8, 10, and 12 months of age time-points as a longitudinal assessment of spatial navigation recall for the P301S and WT mice. Figure 3E demonstrates that the P301S animals had impaired spatial navigation abilities compared to age-matched WT mice at 8 (*p* < 0.01) and 10 months (*p* < 0.05). There was no significant difference at 12 months of age (*p* > 0.05), which was most likely due to the low survival number of the P301S animals at this point due to paralysis and death. Animals that exhibited hindlimb paralysis were not examined in the Barnes maze, and there was no significant difference in total average velocity (mm/s) at the time-points (data not shown). Overall, there was a significant difference in motor performance and spatial learning and memory in longitudinally tested transgenic tau P301S mice from 8 months old that occurred simultaneously with the increase in tau PET imaging signal.

### 3.3. Increased ^18^F-THK5351 PET Signal Coincided with Abundant Tau Brain Pathology

A different set of animals was sacrificed at 6, 8, 10, or 12 months of age to determine the levels of tau pathology in the brain and their correlation with PET imaging and motor, learning, and memory assessment. Histology for tau pathology was performed in P301S and WT animals using AT8 for hyperphosphorylated tau, MC1 for conformational-dependent pre-tangle tau, and classic Gallyas silver staining for NFTs. All histological markers revealed an age-dependent, progressive tau pathology in P301S animals from 6 to 12 months in the hippocampus and cortex (Figure 4A). No histological signal was seen in WT animals at the same ages, as observed in the representative picture.

Following whole brain scan of stained sections, total burden quantification of AT8, MC1, and Gallyas silver staining was performed. There was an increase in total AT8-immunoreactivity burden (*p* = 0.0357, F (3, 13) = 3.855), MC1 burden (*p* = 0.0472, F (3, 12) = 3.57), and Gallyas silver staining burden (*p* = 0.0372, F (3, 12) = 3.898) over time in P301S mice (Figure 4B, one-way ANOVA). Post hoc analysis of AT8 total burden in P301S mice demonstrated a significant difference between 6 and 12 months (*p* < 0.05). Both MC1 immunoreactivity and Gallyas silver staining burden also exhibited significant differences between the 6- and 12-month-old P301S mice (*p* < 0.05, Tukey’s multiple comparisons test).

Another set of P301S mice underwent biochemical studies. Brain tissue was homogenized, and the insoluble (formic acid-soluble) fraction was analyzed by ELISA for pathological tau. The pS199 ELISA (Figure 4C) determined an overall significant effect by one-way ANOVA (*p* = 0.0079, F (3, 20) = 5.23). In addition, there was an age-dependent increase of insoluble pathological tau and elevated fold-change in P301S brain. This increase was significantly different between 6 months and older ages. Compared with 6-month-old mice, the 8-month-old P301S mice displayed a 1.85-fold increase (*p* < 0.05), the 10-month-old P301S mice had a 2.02-fold increase (*p* < 0.01), and the 12-month-old P301S mice showed a 1.74-fold increase (*p* < 0.05). No tau was detected in aged WT littermate brain (data not shown). Thus, concomitant to tau PET imaging and behavior results, there was a progressive and age-related burden of tau pathology in the P301S mouse model. This increase was first observed by biochemical analysis: histological and PET analyses showed significant differences at older ages.

## 4. Discussion

The development of tau PET imaging tracers is critical in order to gain insight into the differences between tau depositing disorders, to diagnose them, and to follow their progression during disease development and treatment trials. Indeed, various molecular probes for tau PET imaging have been developed but still need detailed characterization for sensitivity and specificity in relation to neurodegenerative tauopathies. These agents include ^18^F-THK5105, ^18^F-THK5351, ^18^F-T808, ^18^F-PBB3, ^18^F/^3^H-THK5117, ^18^F-AV1451, and others [14].

Successful Tau PET tracers need to first cross the blood–brain barrier and penetrate the plasma cell membrane since tau is an intracellular protein. Second, Tau PET tracers need to differentiate between Aβ plaques and NFTs. Both are composed of beta-sheet aggregates; hence, tau ligands must have at least a 20–50-fold affinity for tau over Aβ. Third, tau PET tracers must detect tau in its multiple potential conformations, isoforms, and/or post-translational modifications, but differentiate pathogenic from native conformation of tau. Finally, Tau PET tracers must detect early pathological events to have a significant clinical impact [25].

The goal of this study was to determine whether ^18^F-THK5351 demonstrates longitudinal, progressive uptake in a transgenic tau mouse model that anticipates diagnosis to clinical signs or, at least, matches behavioral changes and tau pathology. We found that PET imaging using ^18^F-THK5351 allows for early diagnosis compared to post-mortem histological analysis and it becomes positive simultaneously to changes observed in learning and memory and biochemical alterations.

In vivo tau PET imaging was performed at 6, 8, 10, and 12 months of age in P301S and WT mice. Significant differences in tau PET imaging began at 8 months of age, steadily increased to 10 months of age, and remained significantly augmented at 12 months of age in the overall brain in P301S mice compared to WT rodents. Tau PET imaging matched reduced motor function assessed by rotarod performance in the P301S mice compared to age-matched WT mice, which started at 8 months and was progressive, along with the hunchback posture and hind limb paralysis, as previously reported [18,24]. This age-associated effect is most likely due to the enhanced levels of pathological tau, neuroinflammation, and neuronal loss, as previously denoted in the spinal cord [26]. Tau PET imaging also matched the longitudinal assessment of spatial learning and memory by Barnes maze. This demonstrated impairment in the training phase, originating at 8 months of age and remaining impaired at 10 and 12 months of age in the P301S mice when assessing memory for the escape hole location compared longitudinally to matched WT mice. Moreover, these changes matched those observed with biochemical studies by ELISA of insoluble tau. However, postmortem histological analysis using AT8, MC1, and Gallyas silver staining for tau pathology revealed a significant increase at older ages. The increasing tau pathology in brain over time was observed by motor, cognitive, histological, and biochemical analyses, and this age-dependent effect on tau pathology was proficiently detected by ^18^F-THK5351 PET imaging.

The motor abnormalities described in the P301S mouse model presented some complications to this study. Yoshiyama et al. described about a 20% survival rate by 12 months due to hind limb paralysis [18], which was replicated by our colony in this study (Figure 3B). To this end, there was a lowered *n*-value of P301S animals remaining by the 12-month age-mark. Moreover, surviving P301S mice may exhibit selective differential P301S mutant MAPT expression, as variability in the P301S mouse model has been previously reported as well [27]. This may explain the low values obtained in P301S mice at 12 months when PET imaged. In addition, the P301S mouse model overexpresses human mutant 4R/1N tau. It may be important to test ^18^F-THK5351 PET imaging in other models of tauopathy and tau isoforms, such as a humanized tau mouse that does not develop NFTs [28], or the hTau mouse model that expresses all six human tau isoforms [29]. Moreover, the ^18^F-THK5351 probe did not seem to detect the initial tau deposition in the brain of P301S mice as histological and biochemical assessment does at 6 months. Although, the amount of tau deposition and insoluble levels at 6 months may be considered negligible, detection of these first pathological events could be the key for an early accurate diagnosis. The ^18^F-THK5351 tau tracer, then, may detect NFTs better when greater quantities are present; however, more studies are warranted to test its threshold for detection.

Additional challenges include limitations of small animal PET imaging technology. It is suggested that the reconstructed spatial resolution for small animal PET imaging should be <1 mm in all directions with maximum sensitivity [30]. These constraints could explain why, although elevated ^18^F-THK5351 retention in the whole brain was identified, there was no significance in ROIs, as shown by the spatiotemporal pattern of tau deposition.

Tau has been reported to be composed of an assortment of tau conformations or “strains” represented by different structural heterogeneity or conformational variants and biochemical properties that may result in diverse tauopathies [31,32,33]. It may be that the tau tracer ^18^F-THK5351 is better at detecting a particular tau strain or conformation, as seen in one study of the FTD dementia syndrome named semantic variant, primary progressive aphasia [34]. ^18^F-THK5351 also recognizes distinct topographic patterns in mild cognitive impairment (MCI) patients at risk of AD [35]. Therefore, the tracer may be targeting a strain that is more abundant at later stages than other early conformations. Although this assumption should be further analyzed, it is unclear yet as to what tau conformation(s) triggers the onset of pathology, whether that folding continues during the development of pathology, and what the key strains are that should be identified to accurately diagnose different tauopathies. This is highly relevant when using animal models expressing tau mutations that are unique to some diseases, such as P301S for FTD, but is not present in other disorders, such as CTE or AD, and may not spread equally [36]. Therefore, detection of specific tau strains may be advantageous when imaging patients, but this specificity may be difficult to mimic in animal models.

Although ^18^F-THK5351 tau PET imaging resulted in an overall difference between P301S and WT mice from 8 months of age, post hoc analysis of specified registered brain regions did not show significant differences at analyzed ages. This could have been due to the recent findings that ^18^F-THK5351 may possibly have off-target binding affinity. ^18^F-THK5351 was developed to reduce white matter binding of arylquinoline derivatives, yet compared with ^18^F-THK5117, ^18^F-THK5351 showed higher signal-to-background ratio in in vitro autoradiography experiments with human brain tissue [37]. Previous studies using ^18^F-THK5351 demonstrated that it has a high affinity for monoamine oxidase-B, both in vivo and in vitro [16,17]. This could explain the lack of specificity of the tracer in certain brain regions where NFTs are more abundant, as it may be binding to both tau deposits and the associated inflammatory response triggered by soluble tau. Although this may be interpreted as a limitation for selective detection of tau pathology, our results indicate that it provides a strong correlation between tau-related pathology, clinical changes, and imaging. In addition, other tau PET imaging probes, such as 2-pyrrolopyridinylquinoline derivatives, have been recently developed to selectively target tau aggregates in an effort to reduce this off-target binding [38]. Therefore, the use of ^18^F-THK5351 in the clinic still can provide information about tauopathies onset and development, helping to discriminate from other non-tau-related neurological disorders. In fact, ^18^F-THK5351 is successfully used to diagnose AD when used in combination with other biomarkers such as automated brain volumetry on MRI [39] or ^11^C-Pittsburgh compound B [40]. Additionally, a combination of tau tracers may be best for identifying the disease process underlying tauopathies. Furthermore, the use of ^18^F-THK5351 imaging allows for the identification of different MCI patients subtypes and enables the determination of those who are more vulnerable to developing dementia [41] by detecting both tau and astroglial activation. If ^18^F-THK5351 is able to specifically bind to certain isoforms of tau better than other available tau tracers, it may be considered a more suitable diagnostic tool for certain tauopathies such as AD, CBD, or specific subtypes of FTD.

## 5. Conclusions

Overall, longitudinal in vivo tau PET imaging with ^18^F-THK5351 demonstrated tau-related pathology that was progressive and occurred concomitantly with enhanced histological and biochemical levels of tau, as well as motor and cognitive impairment in P301S mice. This study also demonstrated that ^18^F-THK5351 revealed a significant overall increase in tau deposition beginning at 8 months in these mice, which correlated with clinical signs and postmortem data. ^18^F-THK5351 has been reported to bind off-target monoamine oxidase-B. However, the potential of this tracer to also detect associated astroglial activation may be beneficial for better tracking tau-related pathology. Hence, it may be a very useful tracer for detecting neurofibrillary-related pathology, for diagnosing tauopathy disorders, and for following tau-associated pathology during treatment studies as a measure of efficacy in combination with other biomarkers.

## Figures and Tables

**Figure 1 diagnostics-11-01874-f001:**
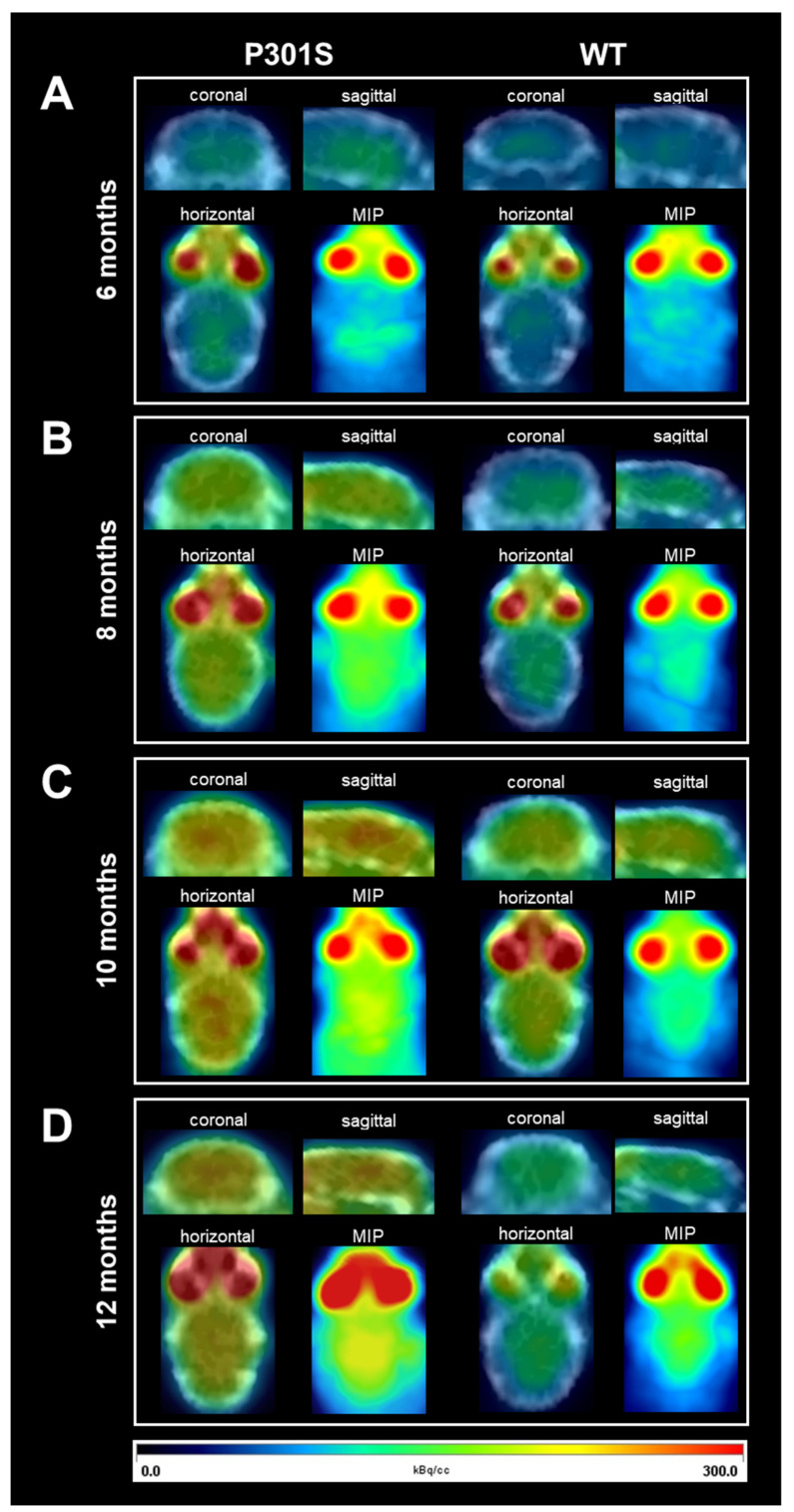
Longitudinal ^18^F-THK5351 tau PET imaging of P301S and WT mice. Illustrative images from positron emission tomography and computerized tomography (PET/CT) imaging in coronal, sagittal, and horizontal views, and maximum intensity projection (MIP) of tau tracer ^18^F-THK5351 uptake in P301S (left) and WT mice (right). Longitudinal analysis was performed at 6 (**A**), 8 (**B**), 10 (**C**), and 12 (**D**) months of age. Images reveal increased PET/CT and MIP signal over time within the same individual, specifically in P301S mice. Color gradient indicates PET signal intensity from black-dark blue (0) to red (300). Color values are indicated in kBq/cc.

**Figure 2 diagnostics-11-01874-f002:**
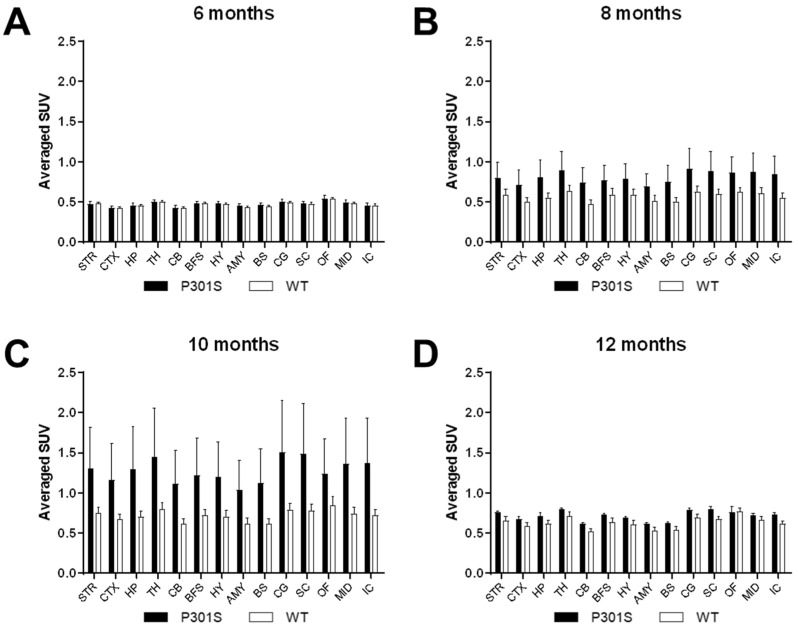
Quantitative analysis of PET imaging. Images were fitted to specific regions of interest (ROIs), and radioligand binding was assessed using averaged standard uptake value (SUV). Quantification was performed in different brain regions in P301S (black bars) and WT (white bars) mice, being plotted and analyzed at 6 (**A**), 8 (**B**), 10 (**C**), and 12 (**D**) months of age. Two-way ANOVA estimated no significant differences at 6 months (*p* > 0.05), whereas analysis of overall brain imaging obtained at 8, 10, and 12 months indicated statistically significant increases (*p* < 0.001, *p* < 0.0001, and *p* < 0.0001, respectively) between P301S and WT mice. STR: striatum, CTX: cortex, HP: hippocampus, TH: thalamus, CB: cerebellum, BFS: basal forebrain and septum, HY: hypothalamus, AMY; amygdala, BS: brain stem, CG; cingulate gyrus, SC: superior colliculi, OF: olfactory bulb, MID: midbrain, IC: inferior colliculi.

**Figure 3 diagnostics-11-01874-f003:**
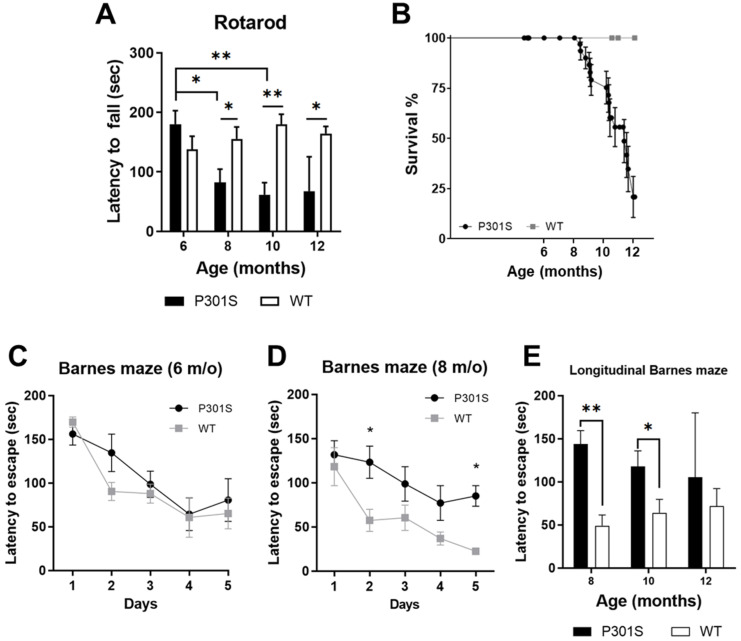
Longitudinal analysis of cognition in P301S and WT animals. (**A**) P301S mice exhibited significant motor impairments by 8 months of age that continued to 12 months of age compared to WT mice. (**B**) P301S mice demonstrated hindlimb paralysis that began at approximately 8.5 months and left about a 20% survival at 12 months of age compared to controls (Mantel–Cox, *p* < 0.0001). (**C**) Barnes maze testing at 6 months of age did not reveal any significant difference in the learning curve of the mice (*p* > 0.05). (**D**) Barnes maze performance at 8 months of age, however, was significantly decreased in P301S mice as revealed by a repeated measures, two-way ANOVA (*p* < 0.001) with post-hoc analysis. (**E**) At 8 and 10 months, the average primary latency on Barnes maze demonstrated that spatial recall was significantly impaired (* *p* < 0.05, ** *p* < 0.01).

**Figure 4 diagnostics-11-01874-f004:**
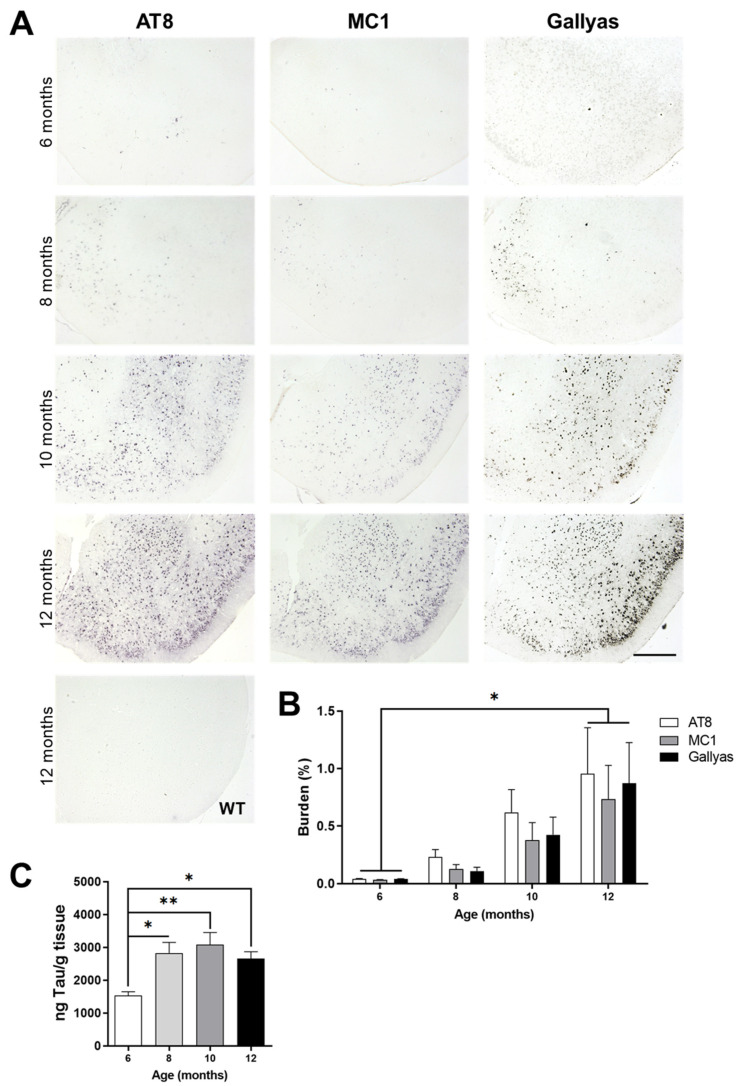
Histological analysis of tau pathology of P301S from 6 to 12 months of age. (**A**) Representative images for AT8, MC1, and Gallyas silver staining for tau pathology in P301S cortex at 6 months, 8 months, 10 months, and 12 months of age. A representative 12-month-old WT brain section stained with AT8 is shown as control. Scale bar: 300 μm. (**B**) Whole brain scan of 4 distinct coronal brain regions revealed a significant, age-dependent difference in quantified AT8 (white bars), MC1 (gray bars), and Gallyas silver staining (black bars) total burden (two-way ANOVA, * *p* < 0.05). (**C**) Fractionated 10% *w/v* brain homogenate revealed a significant difference in insoluble pathological tau levels at 8 or more months of age in P301S mice (one-way ANOVA, Tukey’s post hoc analysis; * *p* < 0.05, ** *p* < 0.01).

## Data Availability

Data supporting reported results will be provided upon request.

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
