# Peer review of "Longitudinal Assessment of Tau-Associated Pathology by ^18^F-THK5351 PET Imaging: A Histological, Biochemical, and Behavioral Study"

_diagnostics, 2021, doi:10.3390/diagnostics11101874_

Round 1

Reviewer 1 Report

Dr. I. Moreno-Gonzalez et al. report regarding the evaluation of 18F-THK5351for detection of hyperphosphorylated tau protein in P301S transgenic mice. This provides interesting insights, however there are several comments on this manuscript as follows.

  1. Throughout the manuscript, the expression of PET probe should provide a uniform style, 18F-THK5351 or [18F]THK5351, including other PET probes.
  2. In Title, “A correlation with histological, biochemical, and behavioral studies” should not be proper, because no correlation analyses among these three parameters are indicated in this manuscript.
  3. In page 4, line 158, the authors describe only injected radioactivity dose as 150 uCi, however the data of specific radioactivity, radiochemical purity, and injected cold compound amount/mouse body weight should be added.
  4. In page 4, line 160, the authors describe the PET data acquisition period as 20 min after the injection. Is 20-min data acquisition proper for quantitative imaging of 18F-THK5351 in mouse brain? Time activity curves in every brain region should be shown.
  5. In page 4, line 160, the authors describe that they measured the ratio of the brain to muscle was calculated. The SUV values in these regions seem to be obtained from over the course of the scan from 0 to 20 min. The authors should evident that this simplified quantification process was proper for 18F-THK5351 analysis.
  6. The authors applied AT8, MC1, and Gallyas staining for histological analyses. However, the explanation what Gallyas is firstly appeared in page 9, line 332, which should be explained in 2.5. Histology for tau pathology at page 4.
  7. In Results part, the authors frequently repeat the descriptions regarding the methods that were already mentioned in 2. Materials and Methods. The Results part should be simpler and clearer.
  8. In Figures 1 and 2, these data seem inconsistent, thus, PET images in Figure 1 demonstrate the increased uptake of 18F-THK5351 into the brains of not only P301S but also WT mice up to 12 months old, however “averaged SUV” in Figure 3 indicated the reduction in 12 months old. Color levels of representative PET images in Figure 1 should be matched to the quantitative data shown in Figure 2.
  9. In Figure 2, the unit of Y-axis is “averaged SUV”. Does it mean “the ratio of the brain to muscle” as pointed out in my comment 5)? If not, which data is“the ratio of the brain to muscle” used?
  10. In Figure 3, the authors report the impaired movement and memory functions in P301S mice compared to WT mice. Are there any relations between these data and the regional brain data obtained using 18F-THK5351-PET measurements shown in Figure 2?
  11. In Figure 4, the staining data of WT mouse should be needed to ensure that the changes in the staining are specifically occurred in P301S mice, not just aging effects.
  12. In Discussion part, in page 11, line 386, the authors described that “it becomes positive in parallel to changes observed in learning and memory and biochemical studies.”, however no such data are indicated in this manuscript. The evidence to support their arguments should be added or delete all such descriptions including the Title.

Author Response

1. Throughout the manuscript, the expression of PET probe should provide a uniform style, 18F-THK5351 or [18F]THK5351, including other PET probes.

All the PET probe have been renamed to 18F-THK5351.

2. In Title, “A correlation with histological, biochemical, and behavioral studies” should not be proper, because no correlation analyses among these three parameters are indicated in this manuscript.

The term “correlation” has been removed from the title.

3. In page 4, line 158, the authors describe only injected radioactivity dose as 150 uCi, however the data of specific radioactivity, radiochemical purity, and injected cold compound amount/mouse body weight should be added.

This information has been included in material and methods section.

4. In page 4, line 160, the authors describe the PET data acquisition period as 20 min after the injection. Is 20-min data acquisition proper for quantitative imaging of 18F-THK5351 in mouse brain? Time activity curves in every brain region should be shown.

After the analysis of the activity curves during dynamic imaging, we observed that SUV reached a plateau between 15 and 20 minutes, therefore, a period of 20 minutes for acquisition was selected to ensure that we were imaging during the right time frame. Animals had a catheter inserted in the tail vein and imaging started right after the administration of the dose and continued for 20 minutes. After this, animals were CT scanned. Materials and methods have been modified to make this information more comprehensible and a supplementary figure has been included to show dynamic acquisition in 6-month-old animals.

5. In page 4, line 160, the authors describe that they measured the ratio of the brain to muscle was calculated. The SUV values in these regions seem to be obtained from over the course of the scan from 0 to 20 min. The authors should evident that this simplified quantification process was proper for 18F-THK5351 analysis.

As indicated in the previous question, SUV plateau was reached around 15 minutes after initial bolus administration, therefore, a period of 20 minutes for acquisition was selected for this tracer.

6. The authors applied AT8, MC1, and Gallyas staining for histological analyses. However, the explanation what Gallyas is firstly appeared in page 9, line 332, which should be explained in 2.5. Histology for tau pathology at page 4.

Explanation of Gallyas staining target has been included in methods next to the other histological processing.

7. In Results part, the authors frequently repeat the descriptions regarding the methods that were already mentioned in 2. Materials and Methods. The Results part should be simpler and clearer.

Results section has been cleaned up and some materials and methods information has been moved to its own section as suggested.

8. In Figures 1 and 2, these data seem inconsistent, thus, PET images in Figure 1 demonstrate the increased uptake of 18F-THK5351 into the brains of not only P301S but also WT mice up to 12 months old, however “averaged SUV” in Figure 3 indicated the reduction in 12 months old. Color levels of representative PET images in Figure 1 should be matched to the quantitative data shown in Figure 2.

Most of the animals that survived over 12 months were WT mice. These animals show SUV values similar to those at 8 months, as observed in figure 1. Only a few transgenic animals survived until 12 months. In figure 1, signal obtained from 12-month P301S animal is very similar to the 8-month-old mouse in PET/CT images but lower than 10 months mice. Although MIP picture looks to have higher signal than at 10 months, final analysis and adjustment based on brain/muscle ratio provides lower value.

9. In Figure 2, the unit of Y-axis is “averaged SUV”. Does it mean “the ratio of the brain to muscle” as pointed out in my comment 5)? If not, which data is “the ratio of the brain to muscle” used?

The averaged SUV obtained in the brain and plotted in figure 2 is already normalized with respect to the muscle signal obtained.

10. In Figure 3, the authors report the impaired movement and memory functions in P301S mice compared to WT mice. Are there any relations between these data and the regional brain data obtained using 18F-THK5351-PET measurements shown in Figure 2?

Movement and memory functions are associated with different brain areas, including motor cortex, cerebellum, midbrain and hippocampus. Since both motor and memory functions are impaired more or less at the same time, and there may be other functions also affected, it may be difficult to directly link results in figures 2 and 3.

11. In Figure 4, the staining data of WT mouse should be needed to ensure that the changes in the staining are specifically occurred in P301S mice, not just aging effects.

WT brain sections were included in the staining and the histological quantification, but they have not been shown in the figures because there was no signal in any of the antibody nor Gallyas staining. A representative picture of a 12-month WT mouse stained with AT8 has now been included in figure 4 to demonstrate this.

12. In Discussion part, in page 11, line 386, the authors described that “it becomes positive in parallel to changes observed in learning and memory and biochemical studies.”, however no such data are indicated in this manuscript. The evidence to support their arguments should be added or delete all such descriptions including the Title.

The analysis performed in this study includes PET imaging, behavior, histology and biochemistry performed in P301S and WT animals at 6, 8, 10 and 12 months of age. Some of these measurements have been performed longitudinally in exactly the same animals, such as PET imaging, memory and motor performance or histology and biochemistry analysis, which was also done on the same animals. Our study was not designed to show causality; however, we do show that all these changes happen at the same time, so there is a correlation. We do not argue for the existence of a link or specific mechanism between them, as we are aware that with the data we have obtained, we cannot draw that conclusion.

Reviewer 2 Report

In this study, authors aim to validate the ability of PET tracer 18F-THK-5351 to detect early progression of tau-dependent pathology and relationships of 18F-THK-5351 signal with pathological signs. Authors showed an increase in 18F-THK-5351 PET signaling over time in transgenic P301S tau mice from 8 months and demonstrated a positive correlation with histological and biochemical tau changes as well as motor, memory, and learning deficits. The study showed that 18F-THK-5351 is a promising PET imaging tracer allowing to detect abnormal tau aggregation and related neuropathology, which, in the future, may contribute to diagnosing the onset of tauopathies, better understanding of their pathogeneses, and developing a reliable biomarker for follow-up monitoring.

Authors should discuss data suggesting 18F-THK-5351 off-target binding to monoamine oxidase-B (MAO-B) as presented by:

- Lerdsirisuk P, Harada R, Hayakawa Y, Shimizu Y, Ishikawa Y, Iwata R, Kudo Y, Okamura N, Furumoto S. Synthesis and evaluation of 2-pyrrolopyridinylquinoline derivatives as selective tau PET tracers for the diagnosis of Alzheimer's disease. Nucl Med Biol. 2021 Feb;93:11-18. doi: 10.1016/j.nucmedbio.2020.10.002. Epub 2020 Oct 26. PMID: 33221641.

To improve the manuscript, authors are recommended to provide information of where PET tracer 18F-THK5351 was obtained/purchased from. The logic of discussion section may be improved by moving the paragraph on study limitations towards the Concussions section. 

Author Response

1. Authors should discuss data suggesting 18F-THK-5351 off-target binding to monoamine oxidase-B (MAO-B) as presented by Lerdsirisuk P, Harada R, Hayakawa Y, Shimizu Y, Ishikawa Y, Iwata R, Kudo Y, Okamura N, Furumoto S. Synthesis and evaluation of 2-pyrrolopyridinylquinoline derivatives as selective tau PET tracers for the diagnosis of Alzheimer's disease. Nucl Med Biol. 2021 Feb;93:11-18. doi: 10.1016/j.nucmedbio.2020.10.002. Epub 2020 Oct 26. PMID: 33221641.

This publication has been included in the discussion section.

2. To improve the manuscript, authors are recommended to provide information of where PET tracer 18F-THK5351 was obtained/purchased from

This information has been included in material and methods section.

3. The logic of discussion section may be improved by moving the paragraph on study limitations towards the Concussions section. 

We moved the limitations paragraph to just before the conclusions to be able to discuss limitations in detail, and included the main limitation of the study in the conclusions, as suggested.

Round 2

Reviewer 1 Report

No further comments.